# The Budyko functions under non-steady state conditions

Roger Moussa[1], Jean-Paul Lhomme[2]

[1]INRA, UMR LISAH, 2 place Viala, 34060 Montpellier, France
[2]IRD, UMR LISAH, 2 place Viala, 34060 Montpellier, France

*Correspondence to*: Roger Moussa (roger.moussa@inra.fr)

**Abstract.** The Budyko functions relate the evaporation ratio $E/P$ ($E$ is evaporation and $P$ precipitation) to the aridity index $\Phi = E_p/P$ ($E_p$ is potential evaporation) and are valid on long timescales under steady state conditions. A new physically based formulation (noted ML) is proposed to extend the Budyko framework under non-steady state conditions taking into account

the change in terrestrial water storage $\Delta S$. The variation in storage amount $\Delta S$ is taken as negative when withdrawn from the area at stake and used for evaporation and positive otherwise, when removed from the precipitation and stored in the area. The ML formulation introduces a dimensionless parameter $H_E = -\Delta S/E_p$ and can be applied with any Budyko function. It represents a generic framework, easy-to-use at various time steps (year, season or month), the only data required being $E_p$, $P$ and $\Delta S$. For the particular case where the Fu-Zhang equation is used, the ML formulation with $\Delta S \leq 0$ is similar to the

analytical solution of Greve et al. (2016) in the standard Budyko space $(E_p/P, E/P)$, a simple relationship existing between their respective parameters. The ML formulation is extended to the space $[(E_p/(P-\Delta S), E/(P-\Delta S)]$ and compared to the formulations of Chen et al. (2013) and Du et al. (2016). The ML (or Greve et al.) feasible domain has similar upper limit to that of Chen et al. and Du et al., but its lower boundary is different. Moreover, the domain of variation of $E_p/(P-\Delta S)$ differs: for $\Delta S \leq 0$ it is bounded by an upper limit $1/H_E$ in the ML formulation, while it is only bounded by a lower limit in Chen et

al.'s and Du et al.'s formulations. The ML formulation can also be conducted using the dimensionless parameter $H_P = -\Delta S/P$ instead of $H_E$, which yields another form of the equations.

## 1 Introduction

The Budyko framework has become a simple tool widely used within the hydrological community to estimate the evaporation ratio $E/P$ at catchment scale ($E$ is evaporation and $P$ precipitation) as a function of the aridity index $\Phi = E_p/P$

($E_p$ is potential evaporation) through simple mathematical formulations $E/P = B_1(\Phi)$ and with long-term averages of the variables. Most of the formulations were empirically obtained (e.g. Oldekop, 1911; Turc, 1954; Tixeront, 1964; Budyko, 1974; Choudhury, 1999; Zhang et al., 2001; Zhou et al., 2015), but some of them were analytically derived from simple physical assumptions (Table 1): (i) the one derived by Mezentsev (1955) and then by Yang et al. (2008), which has the same form as the one initially proposed by Turc (1954) (noted hereafter Turc-Mezentsev); (ii) the one derived by Fu (1981) and

reworked by Zhang et al. (2004) (noted hereafter Fu-Zhang). These two last formulations involve a shape parameter (respectively $\lambda$ and $\omega$), which varies with catchment characteristics and vegetation dynamics (Donohue et al., 2007; Yang et al., 2009; Li et al., 2013; Carmona et al., 2014). When its value increases, actual evaporation gets closer to its maximum rate.

The Budyko framework was initially limited to steady-state conditions on long timescales, under the assumption of negligible change in soil water storage and groundwater. Hydrological processes leading to changes in water storage are not represented and the catchment is considered as closed without any anthropogenic intervention: precipitation is the only input and evaporation and runoff $Q$ the only outputs $(P = E + Q)$. Recently, the Budyko framework has been downscaled to the year (Istanbulluoglu et al., 2012; Wang, 2012; Carmona et al., 2014; Du et al., 2016), the season (Gentine et al., 2012; Chen

et al., 2013; Greve et al., 2016), and the month (Zhang et al., 2008; Du et al., 2016). However, the water storage variation $\Delta S$ cannot be considered as negligible when dealing with these finer timescales or for unclosed basins (e.g. soil, groundwater, reservoir, snow, interbasin water transfer, irrigation; Jaramillo and Destouni, 2015). In these cases, the catchment is considered to be under non-steady state conditions (Fig. 1) and the basin water balance should be written: $P = E + Q + \Delta S$. Table 2 shows some recent formulations of the Budyko framework extended to take into account the change in catchment

water storage $\Delta S$. Chen et al. (2013) (used in Fang et al., 2016) and Du et al. (2016) proposed empirical modifications of the Turc-Mezentsev and Fu-Zhang equations respectively, precipitation $P$ being replaced by the available water supply defined as $(P - \Delta S)$, Du et al. (2016) including the inter-basin water transfer into $\Delta S$. Greve et al. (2016) analytically modified the Fu-Zhang equation in the standard Budyko space $(E_p/P, E/P)$ introducing an additional parameter, whereas Wang and Zhou (2016) proposed in the same Budyko space a formulation issued from the hydrological ABCD model (Alley, 1984), but with

two additional parameters.

The extension of the Budyko framework to non-steady state conditions being a real challenge, this paper aims to propose a new formulation inferred from a clear physical rationale and compared to other non-steady formulations previously derived. The paper is organized as follows. First, we present the new formulation under non-steady state conditions: its upper and lower limits, its generic equations under restricted evaporation in the Budyko space $(E_p/P, E/P)$ and in the space $[E_p/(P-

\Delta S), E/(P-\Delta S)]$. Second, we compare the new formulation to the analytical solution of Greve et al (2016) in the standard Budyko space and to the formulations of Chen et al. (2013) and Du et al. (2016) in the space $[E_p/(P-\Delta S), E/(P-\Delta S)]$.

## 2 New generic formulation under non-steady state conditions

### 2.1 Upper and lower limits of the Budyko framework

In the Budyko framework each catchment is characterized by the three hydrologic variables $P$, $E$ and $E_p$ which are

represented in a 2D space using dimensionless variables equal to the ratio between two of these variables and the third one. In the rest of the paper, following Andréassian et al. (2016), the space defined by $(\Phi = E_p/P, E/P)$ is called "Budyko space"

and the one defined by $(\Phi^1 = P/E_p, E/E_p)$ "Turc space". For steady state conditions $(\Delta S = 0)$ it should be recall that any Budyko function $B_1$ defined in the Budyko space $(E_p/P, E/P)$ generates an equivalent function $B_2$ in the Turc space expressed as:

$$\frac{E}{E_p} = B_2(\Phi^{-1}) = \frac{B_1(\Phi)}{\Phi}, \tag{1}$$

and that any Budyko function verifies the following conditions under steady state conditions: i) $E = 0$ if $P = 0$; ii) $E \leq P$ if $P \leq E_p$ (water limit); iii) $E \leq E_p$ if $P \geq E_p$ (energy limit); iv) $E \rightarrow E_p$ if $P \rightarrow \infty$.

First, we present the upper and lower limits in the Turc space under steady state conditions, when all the water consumed by evaporation comes from the precipitation, the change in water storage $\Delta S$ being nil $(E = P - Q)$. Figure 2a represents the variation of maximum and minimum actual evapotranspiration, respectively $E_x$ and $E_n$, as a function of precipitation $P$ with

dimensionless variables. The upper solid line represents the dimensionless maximum evaporation rate $E_x/E_p$: it follows the precipitation up to $P/E_p = 1$ (the water limit is presented in bold blue on all graphs) and then is limited by potential evaporation $E_x/E_p = 1$ (the energy limit is in bold green). The lower solid line (in bold black) represents the dimensionless minimum evaporation rate $E_n/E_p$ which follows the x-axis: $E_n/E_p = 0$. The feasible domain between the upper and the lower limits is shown in grey. In the Budyko space we have the following relationships: i) when evaporation is maximum, for $E_p/P$

$\leq 1$, $E_x/P = E_p/P$ and for $E_p/P \geq 1$, $E_x/P = 1$; ii) when evaporation is minimum: $E_n/P = 0$. The corresponding Budyko non-dimensional graph is shown in Fig. 2b and represents the upper and lower limits of the feasible domain of $E/P = B_1(E_p/P)$.

Under non-steady state conditions, either a given amount of water $\Delta S$ stored in the area at stake participates to the evaporation process (for instance, groundwater depletion for irrigation), or a given amount of the precipitation $\Delta S$ is stored in the area (soil water, ground water, reservoirs) following the water balance $(E = P - \Delta S - Q)$. As shown in Fig. 1, the storage

amount $\Delta S$ is taken as negative $(\Delta S \leq 0)$ when withdrawn from the area and used for evaporation; it is taken as positive $(\Delta S \geq 0)$ when removed from the precipitation and stored in the area. When $\Delta S$ is negative, $|\Delta S|$ should be lower than $E_p$ because if $|\Delta S| \geq E_p$, evaporation would be systematically equal to $E_p$; and when $\Delta S$ is positive, it should be necessarily lower than $P$. Consequently: $-E_p \leq \Delta S \leq P$. The variable $\Delta S$ is used in a dimensionless form, either as $H_E = -\Delta S/E_p$ or $H_P = -\Delta S/P$, which are positive when additional water is available for evapotranspiration and negative when water is withdrawn from

precipitation. In the following, all the calculations are made with $H_E$ $(-\Phi^{-1} \leq H_E \leq 1)$, but a similar reasoning is conducted using $H_P$ $(-1 \leq H_P \leq \Phi)$ in Appendix A. Taking into account $\Delta S$ makes the upper and lower limits of the feasible domain different.

In the Turc space, the case where evaporation is at its maximum value is visualized as the upper limit in Figs. 2c and 2e (all the available water is used for evaporation). For both cases $\Delta S \leq 0$ (Fig. 2c) or $\Delta S \geq 0$ (Fig. 2e), we have: $E_x = P - \Delta S$ if

$P - \Delta S \leq E_P$ and $E_x = E_P$ if $P - \Delta S \geq E_P$. Written with dimensionless variables, these equations transform into:

$$\text{if } \frac{P}{E_p} \leq 1 + \frac{\Delta S}{E_p} \text{ then } \frac{E_x}{E_p} = \frac{P}{E_p} - \frac{\Delta S}{E_p} = \Phi^{-1} + H_E, \tag{2}$$

$$\text{if } \frac{P}{E_p} \geq 1 + \frac{\Delta S}{E_p} \text{ then } \frac{E_x}{E_p} = 1. \tag{3}$$

For the minimal value of evapotranspiration $E_n$, we have to distinguish two cases depending if $\Delta S \leq 0$ (Fig. 2c) or $\Delta S \geq 0$ (Fig. 2e).

$$if\ \Delta S \leq 0\ then\ \ \frac{E_n}{E_p} = \frac{-\Delta S}{E_p} = H_E\ , \tag{4a}$$

$$if\ \Delta S \geq 0\ then\ \frac{E_n}{E_p} = 0\ . \tag{4b}$$

5       Translating the above equations into the Budyko space (Figs. 2d, f) yields for the upper limits:

$$if\ \ \frac{E_p}{P} \geq \frac{E_p}{E_p + \Delta S}\ \ then\ \ \frac{E_x}{P} = 1 - \frac{\Delta S}{P} = 1 + H_E\frac{E_p}{P} = 1 + H_E\Phi\ , \tag{5}$$

$$if\ \ \frac{E_p}{P} \leq \frac{E_p}{E_p + \Delta S}\ \ then\ \ \frac{E_x}{P} = \frac{E_p}{P} = \Phi\ . \tag{6}$$

Eq. (5) has two limits: when $H_E = 0$, $E_x/P = 1$, and when $\Delta S \to -E_p$ corresponding to $H_E = 1$, $E_x/P \to (1+\Phi)$. For the lower limits in the Budyko space we have:

10   $if\ \Delta S \leq 0\ \ then\ \ \dfrac{E_n}{P} = \dfrac{-\Delta S}{P} = H_E\dfrac{E_p}{P} = H_E\Phi\ ,$       (7a)

$if\ \Delta S \geq 0\ \ then\ \ \dfrac{E_n}{P} = 0\ .$       (7b)

Note that under steady-state conditions, the upper and lower limits are similar in both Turc and Budyko spaces, while this is not the case under non-steady state conditions. It is also interesting to note that for the negative values of $H_E$ the domain of variation of $\Phi$ is bounded $[0, -1/H_E]$ and the possible space of the Budyko functions is limited to a triangle (Fig. 2f).

## 2.2 General equations with restricted evaporation

We examine now the case where the evaporation rate is lower than its maximum possible rate. In the Turc space, under non-steady state conditions ($\Delta S \leq 0$ in Fig. 2c or $\Delta S \geq 0$ in Fig. 2e), Eq. (1) should be transformed to take into account the impact of water storage on the evaporation process. We search a mathematical formulation which transforms the upper and lower

20 limits for the steady state conditions (Fig. 2a) into the corresponding ones for the non-steady state conditions (Fig. 2c if $\Delta S \leq 0$ and Fig. 2e if $\Delta S \geq 0$). The mathematical transformation is searched under the following form $E/E_p = \alpha\ B_2(\Phi^{-1} + \gamma) + \beta$, which combines a x-axis translation ($\gamma$), a y-axis translation ($\beta$) and an homothetic transformation ($\alpha$). This mathematical form is suggested by the way the physical domain of Turc's space is transformed when passing from steady-state conditions to non-steady sate conditions (Figs. 2a, c, e). Note that the reasoning can be conducted either in the Turc or the Budyko

25 space, but the upper and lower limits and the transformation from steady to non-steady state conditions are easier to grasp in the Turc space than in the Budyko space. We distinguish the two cases corresponding to $\Delta S \leq 0$ and $\Delta S \geq 0$.

### 2.2.1 Case $\Delta S \leq 0$

In the Turc space, the lower limit $B_2(\Phi^{-1}) = 0$ in Fig. 2a transforms into $B_2(\Phi^{-1}) = H_E$ in Fig. 2c. Using the mathematical transformation described above, we obtain $(\alpha \times 0) + \beta = H_E$. Following a similar reasoning, the energy limit $B_2(\Phi^{-1}) = 1$ transforms into $B_2(\Phi^{-1}) = 1$, which yields $\alpha + \beta = 1$, and the water limit $B_2(\Phi^{-1}) = \Phi^{-1}$ transforms into $B_2(\Phi^{-1}) = H_E + \Phi^{-1}$, which yields $\alpha (\Phi^{-1} + \gamma) + \beta = H_E + \Phi^{-1}$. The resolution of these three equations gives: $\alpha = 1 - H_E$, $\beta = H_E$ and $\gamma = \Phi^{-1}H_E/(1-H_E)$. Consequently Eq. (1) should be transformed into:

$$\frac{E}{E_p} = (1 - H_E)B_2\left(\frac{\Phi^{-1}}{1-H_E}\right) + H_E .$$ (8)

By introducing Eq. (1) into Eq. (8), we obtain the formulation in the Budyko space (Fig. 2d):

$$\frac{E}{P} = (1 - H_E)\Phi B_2\left(\frac{\Phi^{-1}}{1-H_E}\right) + H_E\Phi = B_1[(1 - H_E)\Phi] + H_E\Phi .$$ (9)

The derivative of Eq. (9) is

$$\frac{d\left(\frac{E}{P}\right)}{d\Phi} = (1 - H_E)\frac{dB_1[(1-H_E)\Phi]}{d\Phi} + H_E .$$ (10)

Given that $\frac{dB_1[(1-H_E)\Phi]}{d\Phi} = 1$ for $\Phi = 0$ and $\frac{dB_1[(1-H_E)\Phi]}{d\Phi} = 0$ when $\Phi \to \infty$, the derivative $\frac{d\left(\frac{E}{P}\right)}{d\Phi}$ (i.e. the slope of the curve) is equal to $1$ for $\Phi = 0$, and when $\Phi \to \infty$, the derivative tends to $H_E$.

### 2.2.2 Case $\Delta S \geq 0$

Following the same reasoning as above, the lower limit, the energy limit and the water limit of $B_2(\Phi^{-1})$ in the Turc space in Fig. 2a (respectively $0$, $1$ and $\Phi^{-1}$) transform respectively into $0$, $1$, and $H_E + \Phi^{-1}$ in Fig. 2e. We obtain respectively the following three equations: $(\alpha \times 0) + \beta = 0$, $\alpha + \beta = 1$ and $\alpha (\Phi^{-1} + \gamma) + \beta = H_E + \Phi^{-1}$. The resolution gives: $\alpha = 1$, $\beta = 0$ and $\gamma = H_E$. Consequently Eq. (1) should be transformed into:

$$\frac{E}{E_p} = B_2(\Phi^{-1} + H_E) .$$ (11)

By introducing Eq. (1) into Eq. (11), we obtain the formulation in the Budyko space (Fig. 2f):

$$\frac{E}{P} = \Phi B_2(\Phi^{-1} + H_E) = (1 + H_E\Phi)B_1\left(\frac{\Phi}{1+H_E\Phi}\right) .$$ (12)

The derivative of Eq. (12) is:

$$\frac{d\left(\frac{E}{P}\right)}{d\Phi} = H_E B_1\left(\frac{\Phi}{1+H_E\Phi}\right) + (1 + H_E\Phi)\frac{d\left[B_1\left(\frac{\Phi}{1+H_E\Phi}\right)\right]}{d\Phi} .$$ (13)

Given that $B_1\left(\frac{\Phi}{1+H_E\Phi}\right) = 0$ and $\frac{d\left[B_1\left(\frac{\Phi}{1+H_E\Phi}\right)\right]}{d\Phi} = 1$ for $\Phi = 0$, the derivative $\frac{d\left(\frac{E}{P}\right)}{d\Phi}$ is equal to $1$ for $\Phi = 0$. When $\Phi \to -1/H_E$, $B_1\left(\frac{\Phi}{1+H_E\Phi}\right) = 1$ and $\frac{d\left[B_1\left(\frac{\Phi}{1+H_E\Phi}\right)\right]}{d\Phi} = 0$, the derivative $\frac{d\left(\frac{E}{P}\right)}{d\Phi}$ tends to $H_E$.

In the following these new generic formulae (Eqs (8) and (9) for $\Delta S \leq 0$ and Eqs (11) and (12) for $\Delta S \geq 0$) are called ML formulation (ML standing for Moussa-Lhomme).

### 2.2.3 Application

Any Budyko equation $B_1(\Phi)$ from Table 1 can be used in Eqs.(9) and (12) as detailed in Table S1 in the "Supplementary material". It is worth noting that both Turc-Mezentsev and Fu-Zhang functions, which are obtained from the resolution of a Pfaffian differential equation, have the following remarkable simple property: $F(1/x) = F(x)/x$. This means that the same mathematical expression is valid for $B_1$ and $B_2$: $B_1 = B_2$. Both Turc-Mezentsev and Fu-Zhang functions have similar shapes, and a simple linear relationship was established by Yang et al. (2008) between their parameters (see Table 1): $\omega = \lambda + 0.72$.

The ML formulation is used hereafter with the Fu-Zhang function (Table 1) for comparison with the analytical solution of Greve et al. (2016) based upon the same function. Replacing $B_1$ by Fu-Zhang's equation, in Eq. (9) for $\Delta S \leq 0$ and in Eq. (12) for $\Delta S \geq 0$, gives in the Budyko space:

$$if\ \Delta S \leq 0\ \ then\ \ \ \frac{E}{P} = 1 + \Phi - [1 + (1 - H_E)^\omega \Phi^\omega]^{\frac{1}{\omega}}, \tag{14a}$$

$$if\ \Delta S \geq 0\ \ then\ \ \ \frac{E}{P} = 1 + (1 + H_E)\Phi - [(1 + H_E\Phi)^\omega + \Phi^\omega]^{\frac{1}{\omega}}. \tag{14b}$$

For $\Phi = 0$, and in both cases $\Delta S \leq 0$ and $\Delta S \geq 0$, we have $E/P = 0$. However the upper limits of $\Phi$ differ: for $\Delta S \leq 0$, when $\Phi \rightarrow \infty$, $E/P \rightarrow \infty$, while for $\Delta S \geq 0$ the maximum value of $\Phi$ is $-1/H_E$ and corresponds to $E/P = 0$. Figure 3 shows some examples of the shape of the ML formulation in the Budyko space (Eqs. 14a, b) for $\omega = 1.5$ and different values of $H_E$. Note that for the particular and unlikely case when $H_E \rightarrow -\infty$, upper and lower limits are reduced to the point $(E_p/P = 0, E/P = 0)$. For $H_E = 0$ we obtain the curves corresponding to steady-state conditions, while for $H_E = 1$, upper and lower limits are superimposed, and the domain is restricted to the 1:1 line. We can easily verify that all functions in Table S1 of the "Supplementary material" give similar results.

### 2.3. The ML formulation in the space $[E_p/(P-\Delta S), E/(P-\Delta S)]$

As mentioned in the introduction, some authors (Chen et al., 2013; Du et al., 2016) have dealt with the non-steady conditions
by modifying the Budyko reference space and replacing the precipitation $P$ by $P-\Delta S$. Hereafter we develop the ML formulations in this new space. The upper limits of the ML formulation can be obtained by transforming Eqs. (5) and (6) defined in the Budyko space. We get respectively:

$$if\ \ \frac{E_p}{P-\Delta S} \geq 1\ \ \ \ \ then\ \ \ \ \ \ \ \ \ \frac{E_x}{P-\Delta S} = 1\ , \tag{15}$$

$$if\ \ \frac{E_p}{P-\Delta S} < 1\ \ \ \ \ then\ \ \ \ \ \ \ \ \ \frac{E_x}{P-\Delta S} = \frac{E_p}{P-\Delta S}\ . \tag{16}$$

The lower limits are obtained from Eqs. (7a, b):

$$if\ \Delta S \leq 0\ then\ \frac{E_n}{P-\Delta S} = \frac{-\Delta S}{P-\Delta S} = H_E \frac{E_p}{P-\Delta S}\,,\tag{17a}$$

$$if\ \Delta S \geq 0\ then\ \frac{E_n}{P-\Delta S} = 0\,.\tag{17b}$$

In the new space, we put:

$$\Phi' = \frac{E_p}{P-\Delta S} = \frac{\Phi}{1+H_E\Phi}\quad or\quad \Phi = \frac{\Phi'}{1-H_E\Phi'}\,.\tag{18}$$

Consequently the relationship between $E/(P-\Delta S)$, $\Phi'$ and $E/P$ is given by

$$\frac{E}{P-\Delta S} = \frac{E}{P}\frac{P}{P-\Delta S} = \frac{1}{1+H_E\Phi}\frac{E}{P} = (1 - H_E\Phi')\frac{E}{P}\,.\tag{19}$$

Inserting Eqs. (9) and (12) into Eq. (19) and expressing $\Phi$ as a function of $\Phi'$ (Eq. 18) lead to the ML formulation in the new space:

$$if\ \Delta S \leq 0\ then\ \frac{E}{P-\Delta S} = \frac{1}{1+H_E\Phi}\{B_1[(1-H_E)\Phi] + H_E\Phi\} = (1 - H_E\Phi')B_1\left[\frac{(1-H_E)\Phi'}{1-H_E\Phi'}\right] + H_E\Phi'\,,\tag{20a}$$

$$if\ \Delta S \geq 0\ then\ \frac{E}{P-\Delta S} = \frac{1}{1+H_E\Phi}(1 + H_E\Phi)B_1\left(\frac{\Phi}{1+H_E\Phi}\right) = B_1(\Phi')\,.\tag{20b}$$

Note that for $\Delta S \geq 0$, $E/(P-\Delta S) = B_1(\Phi')$ is independent of $H_E$ and is identical to the steady state condition $H_E = 0$. This is explained by the fact that, the stored water $\Delta S$ being initially subtracted to the precipitation $P$, it does not participate in the evaporation process and consequently has no impact on the ratio $E/(P-\Delta S)$. For $\Delta S \leq 0$, and for $\Phi = 0$, i.e. $P \to \infty$, we have $\Phi' = 0$, $B_1 = 0$ and $E/(P-\Delta S) = 0$. When $\Phi \to \infty$ which corresponds to $P \to 0$, we have $\Phi' = 1/H_E$, $B_1 = 1$, and $E/(P-\Delta S) \to 1$.

Any Budyko formulation $B_1$ in Table 1 can be used with Eqs. (20a, b), as shown in Table S2 of the "Supplementary material". When the Fu-Zhang equation is used, Eqs. (20a, b) become:

$$if\ \Delta S \leq 0\ then\ \frac{E}{P-\Delta S} = 1 + (1 - H_E)\Phi' - [(1 - H_E\Phi')^\omega + (1 - H_E)^\omega(\Phi')^\omega]^{1/\omega}\,,\tag{21a}$$

$$if\ \Delta S \geq 0\ then\ \frac{E}{P-\Delta S} = 1 + \Phi' - \left(1 + \Phi'^\omega\right)^{\frac{1}{\omega}}\,.\tag{21b}$$

Figure 4 shows the ML formulation (Eqs. 21a, b) in the space $[E_p/(P-\Delta S), E/(P-\Delta S)]$ for $\omega = 1.5$ and different values of $H_E$.
For $H_E = 0$ we retrieve the original Fu-Zhang equation and when $\omega = 1$, we can easily verify that Eqs. (21a, b) are equal to the lower limit of the domain $E/(P-\Delta S) = H_E E_p/(P-\Delta S)$ when $\Delta S \leq 0$, and $E/(P-\Delta S) = 0$ when $\Delta S \geq 0$.

### 2.4. The ML formulation using the dimensionless parameter $H_P$

A mathematical development, similar to the one of Sections 2.1, 2.2 and 2.3, is conducted in Appendix A using the
dimensionless parameter $H_P = -\Delta S/P = H_E\Phi$ (instead of $H_E = -\Delta S/E_P$) and yields another form of the ML formulation. Equivalent mathematical representations are obtained for $\Delta S \leq 0$ and $\Delta S \geq 0$ in the different spaces explored in Sections 2.1, 2.2 and 2.3. In the "Supplementary material", Figs. S1, S2 and S3 obtained with the parameter $H_P$ correspond respectively to Figs. 2, 3 and 4 obtained with $H_E$. Similarly, Tables S3 and S4 (obtained with $H_P$) correspond to Tables S1 and S2 (obtained with $H_E$): they give the ML formulation applied to the different Budyko curves of Table 1 in the standard Budyko space

($E_p/P$, $E/P$) and in the space $[E_p/(P-\Delta S)$, $E/(P-\Delta S)]$. Significant differences appear concerning the mathematical equations and the shape of the feasible domain defined by its upper and lower limits. This is due to the fact that using $H_E$ or $H_P$ corresponds to different sets of data and different functional representations. Both approaches ($H_E$ or $H_P$) can be used. When storage water contributes to enhance evaporation ($\Delta S \leq 0$), $\Delta S$ is bounded by potential evaporation $E_P$ and consequently represents a given percentage of $E_P$. Hence, it is more convenient to use $H_E = -\Delta S/E_p$ instead of $H_P = -\Delta S/P$, because $H_E$ lies in the range *[0, 1]* which is not the case for $H_P$. Conversely, when precipitation water contributes to increase soil water storage ($\Delta S \geq 0$), $\Delta S$ is bounded by $P$ and represents a percentage of precipitation $P$. Consequently, using $H_P$ is more convenient because $H_P$ lies in the range *[-1, 0]*. Moreover, in order to keep the parameter in the range *[0, 1]*, $H'_P = -H_P$ could be preferred.

## 3 Comparing the new formulation with other formulae from the literature

### 3.1 In the standard Budyko space *($E_p/P$, $E/P$)*

When evapotranspiration exceeds precipitation (corresponding herein to the case $\Delta S \leq 0$), Greve et al. (2016) analytically developed a Budyko type equation where the water storage is taken into account through a parameter $y_0$ ($0 \leqslant y_0 \leqslant 1$) introduced into the Fu-Zhang formulation (Table 2). In the Budyko space, this equation writes (Greve et al., 2016; Eq. 9):

$$\frac{E}{P} = 1 + \Phi - [1 + (1 - y_0)^{\kappa-1}\Phi^\kappa]^{1/\kappa} \ . \tag{22}$$

They used the shape parameter $\kappa$ to avoid confusion with the traditional $\omega$ of Fu-Zhang equation. Despite different physical and mathematical backgrounds Eqs. (14a) and (22) are exactly identical and a simple relationship between $H_E$ and $y_0$ can be easily obtained. Equating Eqs. (14a) and (22) with $\omega = \kappa$ yields:

$$H_E = 1 - (1 - y_0)^{\frac{\omega-1}{\omega}} \ . \tag{23}$$

The relationship between $y_0$ and $H_E$ is independent from $\Phi$. It is shown in Fig. 5 for different values of $\omega$. For a given value of $\omega$, we have $H_E < y_0$. For $\omega = 1$, we have $H_E = 0$, and when $\omega \to \infty$ we have $H_E = y_0$.

The derivative of Eq. (22) gives:

$$\frac{d\left(\frac{E}{P}\right)}{d\Phi} = 1 - (1 - y_0)^{\kappa-1}\Phi^{\kappa-1}[1 + (1 - y_0)^{\kappa-1}\Phi^\kappa]^{\frac{1-\kappa}{\kappa}} \ . \tag{24}$$

For $\Phi = 0$ the derivative is equal to *1*, and when $\Phi \to \infty$, the derivative tends to a value noted *m* by Greve et al. (2016; Eq. 12):

$$m = 1 - (1 - y_0)^{\frac{\kappa-1}{\kappa}} \ . \tag{25}$$

The value of the derivative (slope of the curve) is the same in both ML and Greve et al.'s formulations: for $\Phi = 0$ the derivative is equal to 1, and when $\Phi \to \infty$ we have $m = H_E$ (assuming $\omega = \kappa$). Greve et al. (2016; Sect. 4) show that $y_0$ is the

maximum value of $m$ reached when $\omega \to \infty$. Hence, substituting in Eq. (22) $y_0$ by its value inferred from Eq. (23) yields an equation identical to that obtained from the ML formulation (Eq. 14a).

Figure 6 compares the ML formulation Eq. (14a) with Greve et al.'s analytical solution Eq. (22) for $\omega = \kappa = 2$ and different values of $y_0$ (0, 0.2, 0.4, 0.6, 0.8 and 1). The corresponding values of $H_E$ (respectively 0, 0.106, 0.225, 0.367, 0.553 and 1) are calculated using Eq. (23). The new ML formulation with $\omega = \kappa$, and only for $\Delta S \leq 0$, gives exactly the same curves as those obtained by Greve et al. (2016). Both formulations are identical and have the same upper and lower limits. Greve et al. (2016), however, did not mention the lower limit and limited the reasoning to positive values of $y_0$. Moreover, the case $\Delta S \geq 0$ is not considered by Greve et al. (2016).

## 3.2 In the space $[E_p/(P-\Delta S), E/(P-\Delta S)]$

The formulations proposed by Chen et al. (2013) and Du et al. (2016) in the space $[E_p/(P-\Delta S), (E/(P-\Delta S)]$ are essentially empirical. Chen et al. (2013) function (Table 2) is derived from the Turc-Mezentsev equation and written as:

$$\frac{E}{P-\Delta S} = \left[1 + \left(\frac{E_p}{P-\Delta S} - \Phi_t\right)^{-\lambda}\right]^{-\frac{1}{\lambda}} . \tag{26}$$

An additional parameter $\Phi_t$ is empirically introduced in order "to characterize the possible non-zero lower bound of the seasonal aridity index"; this parameter causes a shift of the curve $E/(P-\Delta S)$ along the horizontal axis such as for $E_p/(P-\Delta S) = \Phi_t$ we have $E/(P-\Delta S) = 0$. The derivative of Eq. (26) when $E_p/(P-\Delta S) \to \infty$ is equal to $0$. Similarly, Du et al. (2016) function (Table 2) is an empirical modification of Fu-Zhang equation (Fu, 1981; Zhang et al., 2004) written as:

$$\frac{E}{P-\Delta S} = 1 + \frac{E_p}{P-\Delta S} - \left[1 + \left(\frac{E_p}{P-\Delta S}\right)^{\omega} + \mu\right]^{\frac{1}{\omega}} . \tag{27}$$

A supplementary parameter, noted here $\mu$ (> -1), is added to modify the lower bound of the aridity index $E_P/(P-\Delta S)$. The parameter $\mu$ plays a similar role as $\Phi_t$ in Eq. (26). For $\mu = 0$, Eq. (27) takes the original form of Fu-Zhang equation, $(P-\Delta S)$ replacing $P$. When $\mu$ becomes positive, the lower end of the curve $E/(P-\Delta S)$ shifts to the right. The function $E/(P-\Delta S)$ in Eq. (27) is equal to zero for the particular value of $E_p/(P-\Delta S) = \Phi_d$ such as

$$(1 + \Phi_d)^{\omega} = 1 + (\Phi_d)^{\omega} + \mu . \tag{28}$$

Greve et al.'s formulation can be also written in the space $[E_p/(P-\Delta S), E/(P-\Delta S)]$. Inserting Eq. (22) into Eq. (20a) and expressing $\Phi$ as a function of $\Phi'$ (Eq. 18) leads to:

$$\frac{E}{P-\Delta S} = 1 + (1 - H_E)\Phi' - [(1 - H_E\Phi')^{\kappa} + (1 - y_0)^{\kappa-1}(\Phi')^{\kappa}]^{1/\kappa} . \tag{29}$$

It can be mathematically shown that expressing $(1 - y_0)$ in Eq. (29) as a function of $H_E$ by inverting Eq. (23) (assuming $\omega = \kappa$) leads to the exact ML formulation of Eq. (21a). It is a direct consequence of the similarity of both formulations. Therefore, similar curves to those shown in Fig. 4 for the ML formulation with $H_E \geq 0$ are obtained with Greve et al.'s formulation.

For $\Delta S \geq 0$ (corresponding to $H_E \leq 0$, Fig. 4), the three formulations (Chen et al.'s, Du et al.'s and ML) have similar upper and lower limits. For $\Delta S \leq 0$, Fig. 7 shows an example of the curves obtained with Du et al.'s equation ($\omega = 1.5$) and Chen et al.'s equation with $\lambda = 0.78$ (such as $\lambda = \omega - 0.72$ from Yang et al. (2008)) and with $\Phi_t = \Phi_d = 0.5$ (corresponding to $\mu = 0.484$ from Eq. (28)). Both Chen et al.'s and Du et al.'s formulations are compared to the ML formulation using Fu-Zhang Eq. (14a) with $H_E = +0.25$. The ML and Greve et al.'s formulations are exactly identical if $\kappa = \omega = 1.5$ and $y_0 = 0.578$ calculated from Eq. (23) for $H_E = +0.25$. The four formulations have similar upper limits but the lower limits are different. Both Chen et al.'s and Du et al.'s formulations have the x-axis as lower limit and $E/(P-\Delta S)$ tends to $1$ when $\Phi' = E_p/(P-\Delta S) \rightarrow \infty$, while in the ML formulation with $\Delta S \leq 0$ (Fig. 2a) the feasible domain is a triangle, the domain of variation of $\Phi'$ being limited by $0$ and $1/H_E$.

## 3.3 Discussion

All four formulations, ML, Greve et al. (2016), Chen et al. (2013) and Du et al. (2016), have two parameters each, one for the shape of the curve and another for its shift due to non-steady conditions: $\omega$ and $H_E$ for the ML formulation (with the Fu-Zhang function), $\kappa$ and $y_0$ for Greve et al. (2016), $\lambda$ and $\Phi_t$ for Chen et al. (2013), $\omega$ and $\mu$ for Du et al. (2016). If $H_E = y_0 = \Phi_t = \mu = 0$, the four formulations are identical. For $\Delta S \leq 0$, the ML formulation with Fu-Zhang equation (Eq. 14a) is identical to the one of Greve et al. (2016) in the Budyko space and also in the $[E_p/(P-\Delta S), E/(P-\Delta S)]$ space, provided the shape parameters are assumed to be identical ($\omega = \kappa$) (a simple relationship is established between $H_E$ and the corresponding parameter $y_0$). Despite similar upper limits, the ML and Greve et al. formulations behave very differently from those of Chen et al. and Du et al. in the space $[E_p/(P-\Delta S), E/(P-\Delta S)]$. The ML formulation is different for $\Delta S \leq 0$ and $\Delta S \geq 0$, while those of Chen et al.'s and Du et al.'s do not distinguish the two cases $\Delta S \leq 0$ and $\Delta S \geq 0$. All the formulations have the same upper limits, but the domain of variation of $\Phi'$ differs: respectively $[0, 1/H_E]$ when $\Delta S \leq 0$ and $[0, \infty]$ when $\Delta S \geq 0$ for the ML formulation, $[\Phi_t, \infty]$ for Chen et al. and $[\Phi_d, \infty]$ for Du et al. The lower end of the curve $E/(P-\Delta S)$ corresponds respectively to $(0, 0)$, $(\Phi_t, 0)$ and $(\Phi_d, 0)$ and the upper end to $(1/H_E, 1)$ when $\Delta S \leq 0$ and $(\infty, 1)$ when $\Delta S \geq 0$ for the ML formulation, $(\infty, 1)$ for the other two. Moreover, the ML formulation for $\Delta S \geq 0$ is reduced to a simple relationship $E/(P-\Delta S) = B_1(\Phi')$ and is independent of $H_E$.

It is worth noting that for $\Delta S \leq 0$ the limits of Chen et al. (2013) and Du et al. (2016) functions are not completely sound from a strict physical standpoint: for very high precipitation, when $P >> E_p$, $\Phi$ and $\Phi'$ should logically tend to zero and not to $\Phi_t$ and $\Phi_d$; similarly, when $P \rightarrow 0$, i.e., $\Phi \rightarrow \infty$, it is physically logical that $\Phi' \rightarrow E_p/(-\Delta S)=1/H_E$, as predicted by our Eq. (20a). This tends to prove that the ML formulation, corroborated by Greve et al. (2016) formulation, is physically more correct. Additionally, at simple glimpse, we note that the ML curves could be easily adjusted to the set of experimental points shown in Chen et al. (2013; Figs. 2 and 9) and in Du et al. (2016; Figs. 8 and 9).

# 4 Conclusion

The ML formulations constitute a general mathematical framework which allows any standard Budyko function developed at catchment scale under steady-state conditions (Table 1) to be extended to non-steady conditions (Table S1 in the "Supplementary material"). They take into account the change in catchment water storage $\Delta S$ through a dimensionless parameter $H_E = -\Delta S/E_p$ and the formulation differs according to the sign of $\Delta S$ (Eqs (8) and (9) for $\Delta S \leq 0$ and Eqs (11) and (12) for $\Delta S \geq 0$). Applications can be conducted at various time steps (yearly, seasonal or monthly) both in the Turc space $(P/E_p, E/E_p)$ and in the standard Budyko space $(E_p/P, E/P)$, the only data required to obtain $E$ being $E_p$, $P$ and $\Delta S$.

The new formulations are inferred from an evaluation of the feasible domain of evaporation in the Turc space, adjusted for the case where additional $(\Delta S \leq 0)$ or restricted $(\Delta S \geq 0)$ water is available for evaporation, and then transformed in the Budyko space. For $\Delta S = 0$, the ML formulations return the original equations under steady state conditions, with similar upper and lower limits in both spaces. Under non-steady state conditions, however, the upper and lower limits of the feasible domain differ when using either the Turc or the Budyko space. The ML formulations can be extended to the $[E_p/(P-\Delta S), E/(P-\Delta S)]$ space (Eqs. 20a, b, Fig. 4). They can also be conducted using the dimensionless parameter $H_P = -\Delta S/P$ instead of $H_E$, which yields another form of the equations (Appendix A and "Supplementary material"). It is shown that the ML formulation with $\Delta S \leq 0$ is identical to the analytical solution of Greve et al. (2016) in the standard Budyko space, a simple relationship existing between their respective parameters. On the other hand, the new formulation differs from those of Chen et al. (2013) and Du et al. (2016) in the space $[E_p/(P-\Delta S), E/(P-\Delta S)]$.

# 5 List of symbols

$B_1(\Phi)$     relationship between $E/P$ and $\Phi$ in the Budyko space $(E_p/P, E/P)$ such as $E/P = B_1(\Phi)$ [-].

$B_2(\Phi^{-1})$     relationship between $E/E_p$ and $\Phi^{-1} = P/E_p$ in the Turc space $(P/E_p, E/E_p)$ such as $E/E_p = B_2(P/E_p)$ [-].

$E$     actual evaporation [$LT^{-1}$].

$E_n$     lower limit of the feasible domain of evaporation [$LT^{-1}$].

$E_p$     potential evaporation [$LT^{-1}$].

$E_x$     upper limit of the feasible domain of evaporation [$LT^{-1}$].

$H_E$     $= -\Delta S/E_p (-P/E_p \leq H_E \leq 1)$ [-].

$H_P$     $= -\Delta S/P (-1 \leq H_P \leq E_p/P)$ [-].

$m$     slope of the equation of Greve et al. (2016) when $\Phi \to \infty$ [-].

$ML$     new formulation Eqs (8) and (9) for $\Delta S \leq 0$ and Eqs (11) and (12) for $\Delta S \geq 0$ (stands for Moussa-Lhomme)

$P$     precipitation [$LT^{-1}$].

$Q$     runoff [$LT^{-1}$].

$y_0$     parameter in the Greve et al. (2016) equation accounting for non-steady state conditions ($0 \leq y_0 \leq 1$) [-].

$\kappa$     shape parameter in the Greve et al. (2016) equation corresponding to $\omega$ in the Fu-Zhang equation [-].

$\Delta S$     water storage variation [$LT^{-1}$].

$\lambda$     shape parameter in the Turc-Mezentsev equation ($\lambda > 0$) [-].

$\mu$     parameter in the Du et al. (2013) equation [-].

$\Phi$     aridity index ($\Phi = E_p/P$) [-].

$\Phi_d$     aridity index threshold in the Du et al. (2016) equation corresponding to $E/(P\text{-}\Delta S) = 0$ [-].

$\Phi_t$     aridity index threshold in the Chen et al. (2013) equation [-].

$\Phi'$     $= E_p/(P\text{-}\Delta S)$  [-].

$\omega$     shape parameter of the Fu-Zhang equation ($\omega > 1$) [-].

## Appendix A: Scaling $\Delta S$ by $P$ instead of $E_p$

The Appendix A presents the set of equations when scaling the change in soil water storage $\Delta S$ by precipitation $P$ instead of potential evaporation $E_p$, i.e., using $H_P = -\Delta S/P = H_E \Phi$ ($-1 \leq H_P \leq \Phi$) instead of $H_E = -\Delta S/E_p$ ($-\Phi^{-1} \leq H_E \leq 1$).

### A.1 Upper and lower limits of the Budyko framework

In the Turc space, the upper limits of evapotranspiration $E_x/E_p$ are obtained from Eqs (2 and 3):

$$if \ \frac{P}{E_p} \leq 1 + \frac{\Delta S}{E_p} \qquad then \qquad \frac{E_x}{E_p} = \frac{P}{E_p} - \frac{\Delta S}{E_p} = (1 + H_P)\Phi^{-1}, \tag{A1}$$

$$if \ \frac{P}{E_p} \geq 1 + \frac{\Delta S}{E_p} \qquad then \qquad \frac{E_x}{E_p} = 1 \ , \tag{A2}$$

and the lower limits of evapotranspiration $E_n/E_p$ from Eqs (4a, b):

$$if \ \Delta S \leq 0 \ then \ \ \frac{E_n}{E_p} = -\frac{\Delta S}{E_p} = H_P \Phi^{-1}, \tag{A3a}$$

$$if \ \Delta S \geq 0 \ then \ \frac{E_n}{E_p} = 0 \ . \tag{A3b}$$

The translation in the Budyko space yields for the upper limits:

$$if \ \frac{E_p}{P} \geq \frac{E_p}{E_p + \Delta S} \qquad then \qquad \frac{E_x}{P} = 1 - \frac{\Delta S}{P} = 1 + H_P \ , \tag{A4}$$

$$if \ \frac{E_p}{P} \leq \frac{E_p}{E_p + \Delta S} \qquad then \qquad \frac{E_x}{P} = \frac{E_p}{P} = \Phi \ , \tag{A5}$$

and for the lower limits:

$$if \ \Delta S \leq 0 \ then \ \ \frac{E_n}{P} = -\frac{\Delta S}{P} = H_E \frac{E_p}{P} = H_E \Phi = H_P \ , \tag{A6a}$$

$$if \ \Delta S \geq 0 \ then \ \frac{E_n}{P} = 0 \ . \tag{A6b}$$

In the "Supplementary material", Fig. S1 shows the upper and lower limits of the feasible domain of evaporation in the Turc and Budyko spaces, drawn with the parameter $H_P = -\Delta S/P$. Figure S1 corresponds to Fig. 2 obtained with $H_E = -\Delta S/E_p$.

**A.2 General equations with restricted evaporation**

We distinguish two cases: $\Delta S \leq 0$ and $\Delta S \geq 0$. Substituting $H_E$ by $H_P/\Phi$ in Eqs (8, 9, 11 and 12) we obtain:

If $\Delta S \leq 0$

in the Turc space: $\frac{E}{E_p} = (1 - H_P\Phi^{-1})B_2\left(\frac{\Phi^{-1}}{1-H_P\Phi^{-1}}\right) + H_P\Phi^{-1}$ ,      (A7)

in the Budyko space: $\frac{E}{P} = B_1(\Phi - H_P) + H_P$ .      (A8)

If $\Delta S \geq 0$

in the Turc space: $\frac{E}{E_p} = B_2[(1 + H_P)\Phi^{-1}]$ ,      (A9)

in the Budyko space: $\frac{E}{P} = (1 + H_P)B_1\left(\frac{\Phi}{1+H_P}\right)$ .      (A10)

     Replacing $B_1$ by Fu-Zhang's equation, in Eq. (A8) for $\Delta S \leq 0$ and in Eq. (A10) for $\Delta S \geq 0$, gives in the Budyko space:

*if $\Delta S \leq 0$ then*    $\frac{E}{P} = 1 + \Phi - [1 + (\Phi - H_P)^\omega]^{\frac{1}{\omega}}$ ,      (A11a)

*if $\Delta S \geq 0$ then*    $\frac{E}{P} = 1 + \Phi + H_P - [(1 + H_P)^\omega + \Phi^\omega]^{\frac{1}{\omega}}$ .      (A11b)

In the "Supplementary material", Fig. S2 shows an example of the ML formulation (Eqs. A11a, b) in the Budyko space
obtained with the parameter $H_P = -\Delta S/P$. It corresponds to Fig. 3 obtained with $H_E = -\Delta S/E_p$. Table S3 gives the ML formulation applied to the different Budyko curves of Table 1 with the parameter $H_P$ (Eqs. A8 and A10). It corresponds to Table S1 obtained with $H_E$.

**A.3 The ML formulation in the space [$E_p/(P-\Delta S)$, $E/(P-\Delta S)$]**

Eqs (15, 16, 17a and b) yield for the upper limits:

*if*   $\frac{E_p}{P-\Delta S} \geq 1$     *then*       $\frac{E_x}{P-\Delta S} = 1$ ,      (A12)

*if*   $\frac{E_p}{P-\Delta S} \leq 1$     *then*       $\frac{E_x}{P-\Delta S} = \frac{E_p}{P-\Delta S}$ ,      (A13)

and for the lower limits:

*if $\Delta S \leq 0$*   *then*   $\frac{E_n}{P-\Delta S} = \frac{-\Delta S}{P-\Delta S} = H_E\frac{E_p}{P-\Delta S} = \frac{H_P}{H_P+1}$ ,      (A14a)

*if $\Delta S \geq 0$ then*   $\frac{E_n}{P-\Delta S} = 0$ .      (A14b)

In the new space *[$E_p/(P-\Delta S)$, $E/(P-\Delta S)$]*, we put:

$$\Phi' = \frac{E_p}{P-\Delta S} = \frac{\Phi}{1+H_P} \quad \text{or} \quad \Phi = (1+H_P)\Phi' . \tag{A15}$$

Consequently the relationship between $E/(P-\Delta S)$ and $E/P$ is given by:

$$\frac{E}{P-\Delta S} = \frac{E}{P}\frac{P}{P-\Delta S} = \frac{1}{1+H_p}\frac{E}{P} . \tag{A16}$$

Replacing $H_E$ by $H_P/\Phi$ in Eqs (20a, b) we obtain:

$$if\ \Delta S \leq 0\ then\ \ \frac{E}{P-\Delta S} = \left(\frac{1}{1+H_P}\right)[B_1(\Phi - H_P) + H_P] = \frac{1}{1+H_P}B_1[(1+H_P)\Phi' - H_P] + \frac{H_P}{1+H_P} , \tag{A17a}$$

$$if\ \Delta S \geq 0\ then\ \ \frac{E}{P-\Delta S} = \left(\frac{1}{1+H_P}\right)(1+H_P)B_1\left(\frac{\Phi}{1+H_P}\right) = B_1(\Phi') . \tag{A17b}$$

Using the Fu-Zhang equation for $B_1$ we get:

$$if\ \Delta S \leq 0\ then\ \ \frac{E}{P-\Delta S} = 1 + \Phi' - \frac{H_P}{1+H_P} - \left[\left(\frac{1}{1+H_P}\right)^\omega + \left(\Phi' - \frac{H_P}{1+H_P}\right)^\omega\right]^{1/\omega} , \tag{A18a}$$

$$if\ \Delta S \geq 0\ then\ \ \frac{E}{P-\Delta S} = 1 + \Phi' - \left(1 + \Phi'^\omega\right)^{\frac{1}{\omega}}. \tag{A18b}$$

In the "Supplementary material", Fig. S3 shows an example of the ML formulation (Eqs. A18a, b) in the space *[E$_p$/(P-ΔS), E/(P-ΔS)]* obtained with the parameter $H_P = -\Delta S/P$. It corresponds to Fig. 4 obtained with $H_E = -\Delta S/E_p$. Table S4 gives the ML formulation applied to the different Budyko curves of Table 1 in the space *[E$_p$/(P-ΔS), E/(P-ΔS)]* with the parameter $H_P$ (Eqs. A17a and A17b). It corresponds to Table S2 obtained with $H_E$.

**Acknowledgements**

The authors are very grateful to the reviewers, Drs Gudmundsson and Jaramillo, and to the Editor Dr Coenders-Gerrits, for their constructive comments of the manuscript. They also gratefully acknowledge the scientific and financial support of the UMR LISAH, as well as the always inspiring advice of A. Gaby.

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

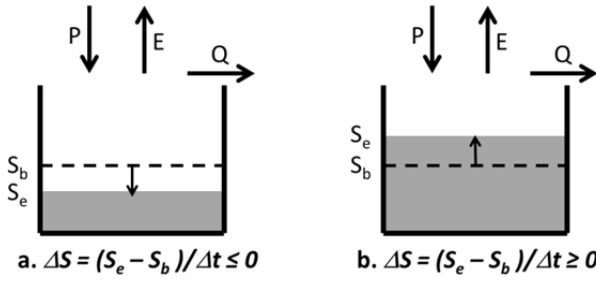

**Figure 1: Representation of the change in soil water storage** $\Delta S = (S_e - S_b)/\Delta t$ **for the two cases considered in the paper:** $\Delta S \leq 0$ **and** $\Delta S \geq 0$. $S_b$ **and** $S_e$ **are respectively the storage at the beginning and the end of the time period** $\Delta t$.

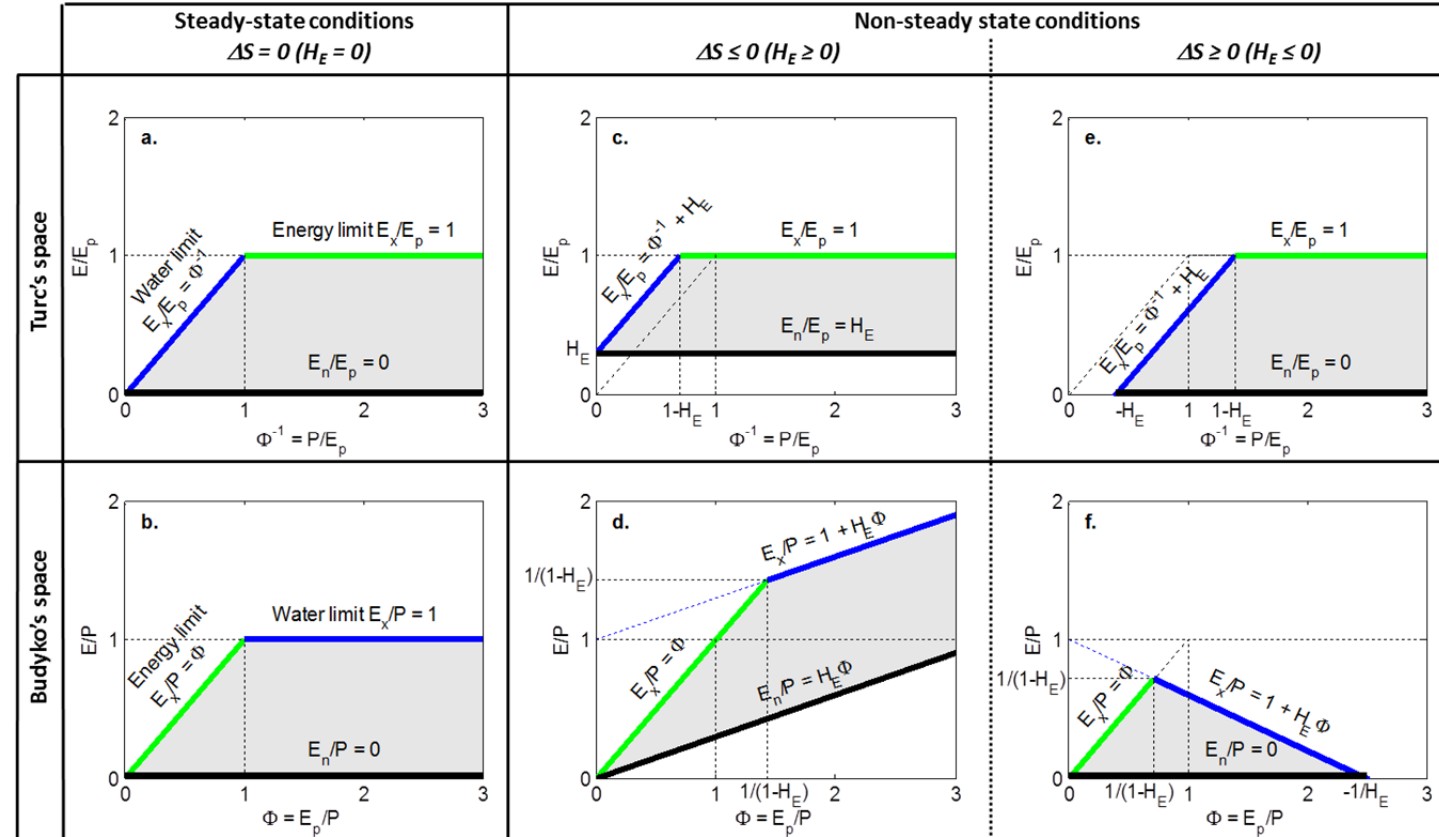

**Figure 2: Upper and lower limits of the feasible domain (in grey) of evaporation in the Turc space** $(P/E_p, E/E_p)$ **and in the Budyko space** $(E_p/P, E/P)$ **(water limit in blue, energy limit in green and lower limit in black) when using the non-dimensional parameter** $H_E$: **(a and b) for steady state conditions; (c, d, e and f) for non-steady state conditions with a storage term** $\Delta S$ **(c and d for** $\Delta S \leq 0$ **and e and f for** $\Delta S \geq 0$).

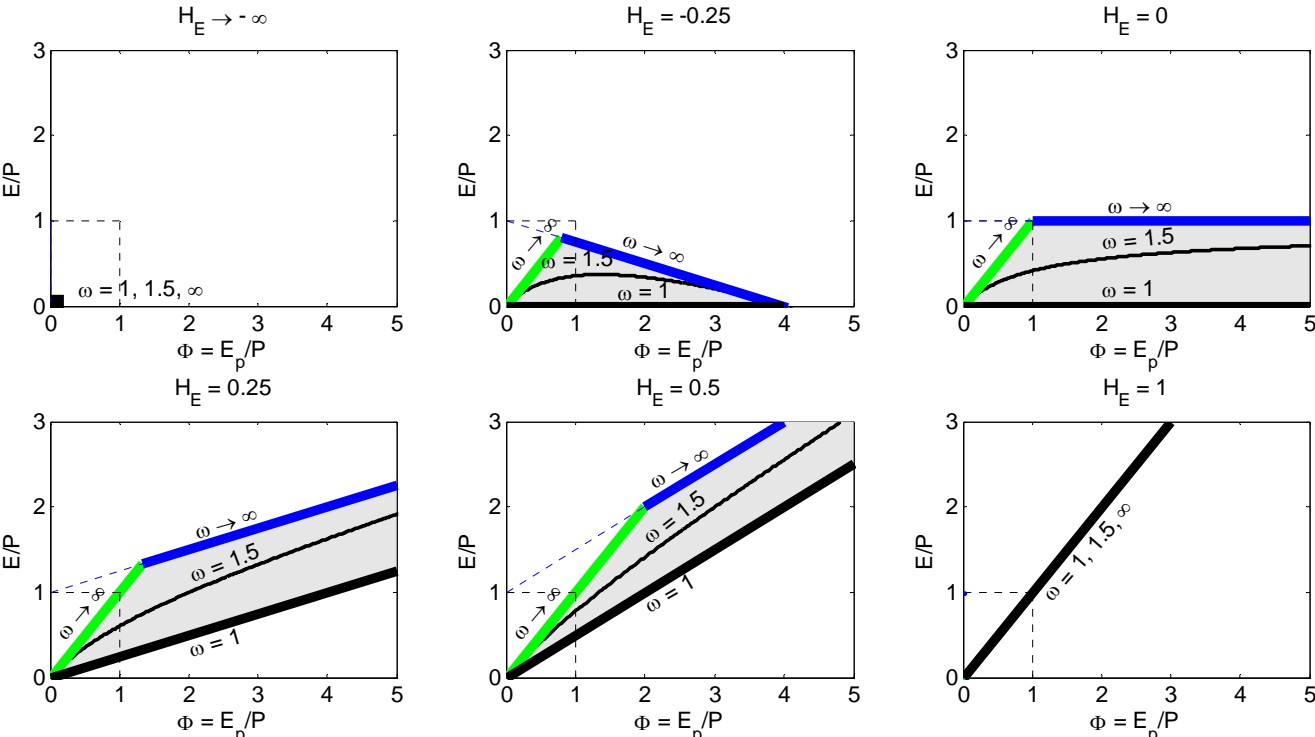

**Figure 3: The ML formulation in the Budyko space with the Fu-Zhang relationship Eqs. (14a, b) for *ω = 1.5* and for different values of *$H_E$.* The bold lines indicate the upper and lower limits of the feasible domain of evaporation shown in grey.**

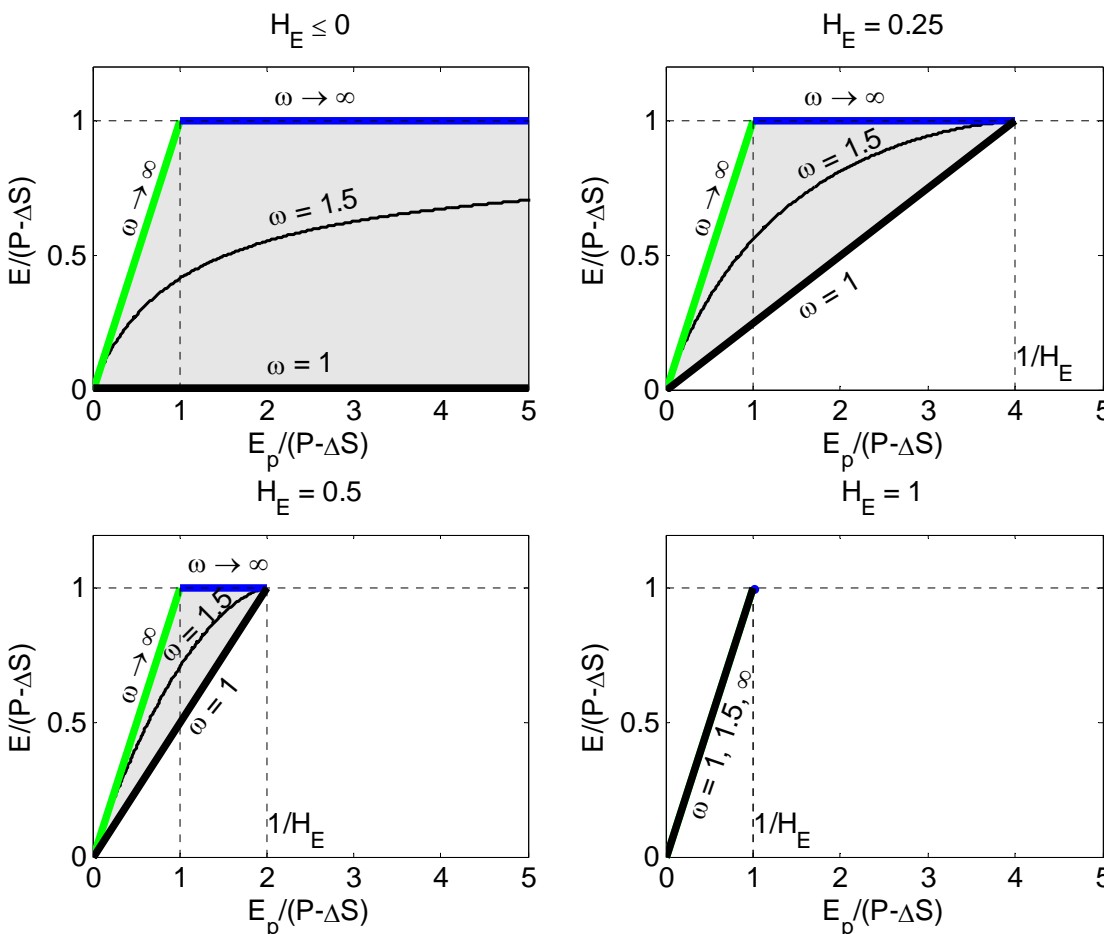

**Figure 4: The ML formulation with the Fu-Zhang Eqs. (21a, b) in the space *[E_p/(P-ΔS), E/(P-ΔS)]* for *ω = 1.5* and four values of *H_E*. For *H_E ≥ 0*, all curves have a common upper end at *Φ' = 1/H_E* corresponding to *E/(P-ΔS) = 1*. The bold lines indicate the upper and lower limits of the feasible domain shown in grey. For *H_E ≤ 0* the curve is similar to the one under steady state conditions.**

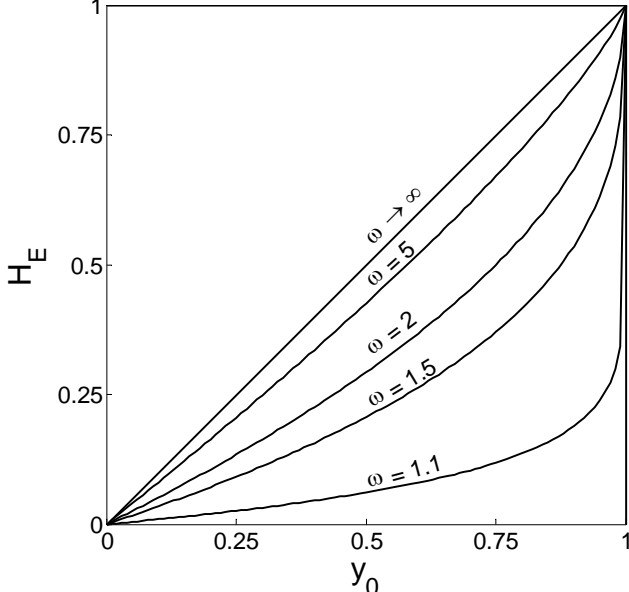

**Figure 5: Relationship (Eq. 23) between the parameter $H_E$ of the ML formulation (Eq. 14a) and the parameter $y_0$ of the Greve et al. (2016) equation (Eq. 22) for different values of $\omega$ with $\omega = \kappa$.**

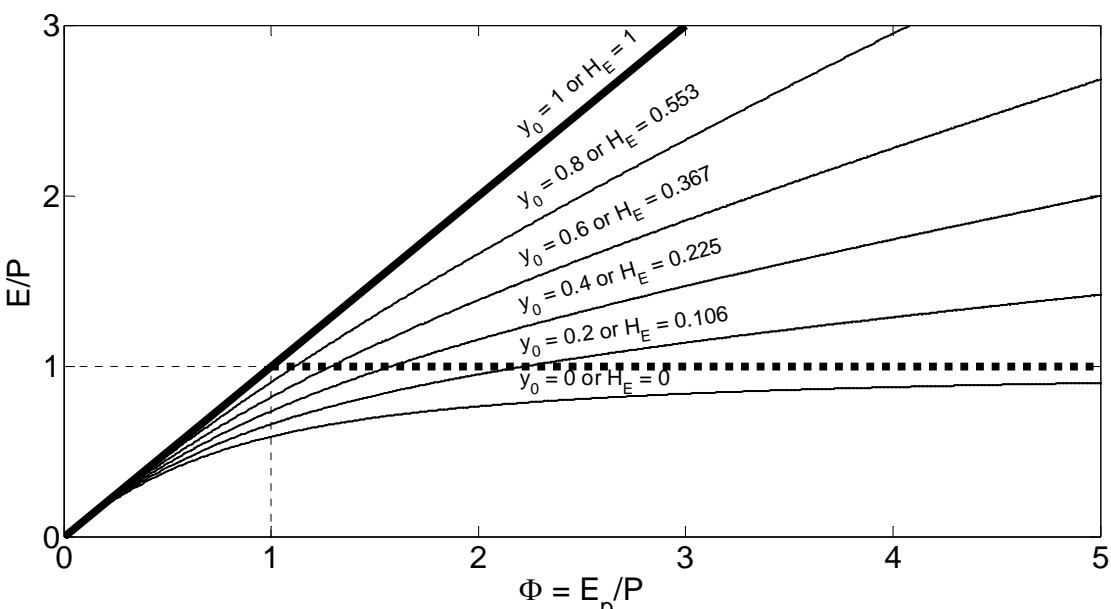

5 **Figure 6: Example showing the similarity of the ML formulation Eq. (14a) and the equation of Greve et al. (2016) Eq. (22) (with $\omega = \kappa = 2$) for different values of $y_0$; the corresponding values of $H_E$ are calculated using Eq. (23).**

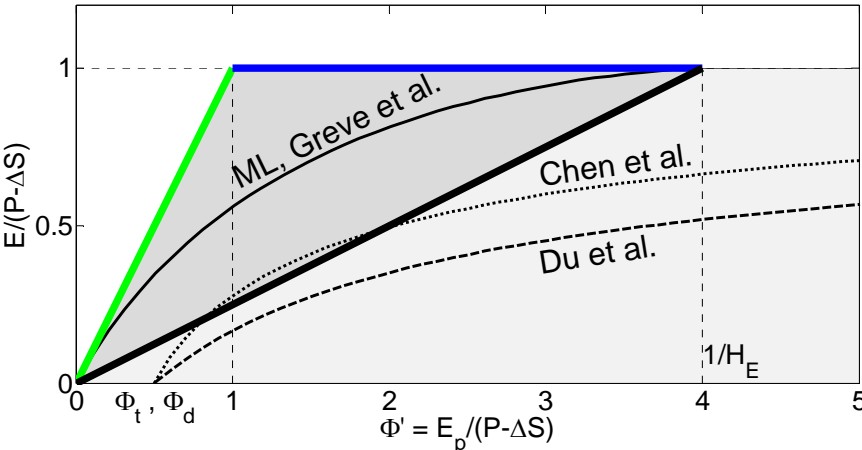

Figure 7: Example comparing in the space *[Ep/(P-ΔS), E/(P-ΔS)]* the three formulations: Du et al. (2016) with *ω = 1.5* and *Φd = 0.5*; Chen et al. (2013) with *λ = ω - 0.72 = 0.78* and *Φt = Φd = 0.5*; the ML formulation for *ΔS ≤ 0* (Eq. 14a) with *ω = 1.5* and *HE = 0.25* (identical to Greve et al. (2016) formulation). The feasible domain of the ML formulation is in dark grey superimposed to the domains of both Chen et al. and Du et al. in light grey.

**Table 1: Different expressions for the Budyko curves under steady state conditions.**

| Reference | Equation $E/P = B_1(\Phi)$ |
|---|---|
| Budyko (1974) | $\dfrac{E}{P} = \left\{ \Phi \tanh \left( \dfrac{1}{\Phi} \right) [1 - \exp(-\Phi)] \right\}^{1/2}$ |
| Turc (1954) with $\lambda = 2$, Mezentsev (1955), Yang et al. (2008) | $\dfrac{E}{P} = \Phi \left( 1 + \Phi^{\lambda} \right)^{-\frac{1}{\lambda}}$ |
| Fu (1981), Zhang et al. (2004) | $\dfrac{E}{P} = 1 + \Phi - (1 + \Phi^{\omega})^{\frac{1}{\omega}}$ |
| Zhang et al. (2001) | $\dfrac{E}{P} = \dfrac{1 + w\Phi}{1 + w\Phi + \Phi^{-1}}$ |
| Zhou et al. (2015) | $\dfrac{E}{P} = \Phi \left( \dfrac{k}{1 + k\Phi^n} \right)^{1/n}$ |

**Table 2: Different expressions for the Budyko curves under non-steady state conditions.**

| Reference | Steady state conditions $B_1(\Phi)$ | Non-steady state conditions |
|---|---|---|
| Greve et al. (2016) | Fu-Zhang | $\dfrac{E}{P} = 1 + \dfrac{E_p}{P} - \left[ 1 + (1 - y_0)^{\kappa - 1} \left( \dfrac{E_p}{P} \right)^{\kappa} \right]^{1/\kappa}$ with $\kappa$ and $y_0$ parameters. |
| Chen et al. (2013) | Turc-Mezentsev | $\dfrac{E}{P\text{-}\Delta S} = \left[ 1 + \left( \dfrac{E_p}{P\text{-}\Delta S} \text{-} \Phi_t \right)^{-\lambda} \right]^{\frac{1}{\lambda}}$ with $\lambda$ and $\Phi_t$ parameters. |
| Du et al. (2016) | Fu-Zhang | $\dfrac{E}{P\text{-}\Delta S} = 1 + \dfrac{E_p}{P\text{-}\Delta S} - \left[ 1 + \left( \dfrac{E_p}{P\text{-}\Delta S} \right)^{\omega} + \mu \right]^{\frac{1}{\omega}}$ with $\omega$ and $\mu$ parameters. |

