# Peer review of "The Budyko functions under non-steady state conditions"

_Hydrology and Earth System Sciences, 2016_

## Referee Comment (RC1) · L. Gudmundsson (Referee) · 3 Aug 2016

The authors present an interesting study in which they propose an approach to extend Budyko functions to non-steady state conditions. The approach is based on a careful evaluation of the feasible limits of the Budyko-Turc space, which are subsequently adjusted for the case where additional water is available for evapotranspiration. This yields a general framework into which common Budyko functions can be inserted. Finally, the authors apply the proposed framework to popular Budyko functions and compare the results to previous studies.

Interestingly, the authors show that if their approach is applied to Fu's equation (Fu, 1981), their approach yields an equation that is mathematically identical to a recent

extension (Greve et al, 2016), after minor rearrangements. This finding increases the credibility of the results as the presented study and Greve et al (2016) have derived the same results on the basis of two independent approaches. Nevertheless, the presented work is clearly a new development as it (1) offers a more general approach that is applicable to a wide range of Budyko functions and (2) provides more explicit insights into the role of water storage (S) to the y0 parameter identified by Greve et al (2016).

Overall the paper is clearly structured and I find the graphical derivation of the proposed extension very convincing. Consequently, I do recommend the publication of the proposed work after some specific comments have been accounted for.

SPECIFIC COMMENTS:

** ** Specific Comment 1: ** **

Although the paper is well structured I had the impression that it would benefit from some linguistic fine tuning and that some sections could presented more clearly.

** ** Specific Comment 2: ** **

The authors mention that the equation that is derived using their approach and Fu's equation, yields an equation that is "similar" to the equation derived by Greve et al (2016). In fact, the two equations are IDENTICAL after some minor re-arrangements, which is also shown by the authors. I therefore would like to urge the authors to clarify this issue in the revised manuscript. (As noted above, the authors work is nevertheless very valuable as it provides an independent validation of the previously derived function and allows for an explicit assessment of the amount of storage water that is available for evapotranspiration).

** ** Specific Comment 3: ** **

Page 4, lines 8ff: This section contains the actual derivation of the authors approach to incorporate storage water into Budyko functions. Unfortunately, I had to read this section several times before I could understand the logic underlying their approach.

Therefore, I would like to encourage the authors to expand this section, and explain the important steps in more detail. More specifically: (1) I was wondering why the authors did search for the equation shown in line 12. (2) It took me a while to figure out how the values of beta, alpha and gamma were chosen (one or two sentences explaining the logic behind this would be helpful).

---

## Referee Comment (RC2) · F. Jaramillo (Referee) · 6 Aug 2016

The authors present a formulation for the use of the Budyko framework for non-steady conditions, i.e., with change in water storage within the basin. I find the manuscript an interesting approach that starts from definitions of water availability and energy demand in the "Turc space", later transposed to the "Budyko space", to end up with a formulation expressing the evaporative ratio in terms of change in storage and aridity index. Advantages: Their non-steady conditions formulation in its final way (Eq. 9) is simple, and can be obtained easily from any other steady-state formulation. It also confirms the robustness of Greve et al. (2016) and finds some important differences with those of Chen et al. (2013) and Du et al. (2013). I also appreciate the literature review on the theory behind the use of the Budyko framework for non-steady conditions. Some suggestions to improve the manuscript are:

1. I find that the start from the "Turc space" and constant change to "Budyko space" gets confusing sometimes. Can't their formulation start directly from the much more commonly used "Budyko space"?

2. I find the term S* somehow difficult to grasp. First, why not just use DeltaS, for better clarity, instead of S=-delta S? Second, why not divide DeltaS (water) by P (water) instead of by EP (energy)? This would make much more sense, expressing the change in storage relative to P, something like S*=DeltaS/P. I think in this way it would be so straightforward to use by anyone...

3. The S limit definition of Line 12 page 3: 0<S<Ep, can the authors then explain in more detail this S limit definition (Line 12 page 3) for clarity? This because as it is, S is always positive, implying that delta S is always negative. So what about water storage in reservoirs (delta S >0), could the ML formulation for non-steady conditions also be used to represent this condition? Or if there is a typo there, could the ML formulation be applied conversely, deltaS<0, e.g. groundwater depletion for irrigation? See definition for both cases in "Local flow regulation and irrigation raise global human water consumption and footprint", 2015, Supplementary Information.

4. Upgrade the justification of their study (Line 20-21, page 2), something like a very-well needed validation, integration and comparison of non-steady formulations in Budyko space; that is what their work is from my point of view?

5. Why would I prefer the ML formulation, please expand? I think the fact that no-additional parameters other than PET, P and deltaS to obtain ET/P for non-steady conditions is an important advantage.

6. One of the main conclusions is 25-28 page 8: Just by reading the corresponding discussion (Line 6-14, page 8) it is somehow difficult to understand. Can the authors

use an additional figure comparing for the same storage conditions ALL the four formulations, Greve et al., ML, Chen et al. and Du et al.], either in the normal Budyko [Ep/P, E/P] or in the modified space [Ep/(P+dS), E/(P+dS)]. This synthesis would be very helpful for the reader and potential users of the ML formulation!

———————————————

---

## Author Comment (AC1) · 14 Sep 2016

**Responses to reviewers**

We are very grateful to both reviewers Dr Gudmundsson and Dr Jaramillo for their constructive comments of the manuscript. We totally agree with all their recommendations.

**Referee #1: L. Gudmundsson**

*The authors present an interesting study in which they propose an approach to extend Budyko functions to non-steady state conditions. The approach is based on a careful evaluation of the feasible limits of the Budyko-Turc space, which are subsequently adjusted for the case where additional water is available for evapotranspiration. This yields a general framework into which common Budyko functions can be inserted. Finally, the authors apply the proposed framework to popular Budyko functions and compare the results to previous studies.*

*Interestingly, the authors show that if their approach is applied to Fu's equation (Fu,1981), their approach yields an equation that is mathematically identical to a recent extension (Greve et al, 2016), after minor rearrangements. This finding increases the credibility of the results as the presented study and Greve et al (2016) have derived the same results on the basis of two independent approaches. Nevertheless, the presented work is clearly a new development as it (1) offers a more general approach that is applicable to a wide range of Budyko functions and (2) provides more explicit insights into the role of water storage (S) to the y0 parameter identified by Greve et al (2016).*

*Overall the paper is clearly structured and I find the graphical derivation of the proposed extension very convincing. Consequently, I do recommend the publication of the proposed work after some specific comments have been accounted for.*

Thank you!

*SPECIFIC COMMENTS:*

*1. ** ** Specific Comment 1: ** ***

*Although the paper is well structured I had the impression that it would benefit from some linguistic fine tuning and that some sections could presented more clearly.*

Ok. Additional comments will be made to clarify some sections, and linguistic refinement to improve the text.

*2. ** ** Specific Comment 2: ** ***

*The authors mention that the equation that is derived using their approach and Fu's equation, yields an equation that is "similar" to the equation derived by Greve et al (2016). In fact, the two equations are IDENTICAL after some minor re-arrangements, which is also shown by the authors. I therefore would like to urge the authors to clarify this issue in the revised manuscript. (As noted above, the authors work is nevertheless very valuable as it provides an*

*independent validation of the previously derived function and allows for an explicit assessment of the amount of storage water that is available for evapotranspiration).*

Ok. A discussion comparing the ML and the Greve et al's approaches will be added in order to clarify how two totally different methods based on very different hypotheses give exactly the same result and equation. We will discuss the relationship between $S^*$ of the ML formulation and $y_0$ of Greve et al.

*3. ** ** Specific Comment 3: ** ***

*Page 4, lines 8ff: This section contains the actual derivation of the authors approach to incorporate storage water into Budyko functions. Unfortunately, I had to read this section several times before I could understand the logic underlying their approach. Therefore, I would like to encourage the authors to expand this section, and explain the important steps in more detail. More specifically: (1) I was wondering why the authors did search for the equation shown in line 12. (2) It took me a while to figure out how the values of beta, alpha and gamma were chosen (one or two sentences explaining the logic behind this would be helpful).*

Ok. This paragraph will be rewritten in order to clarify the choice of the function used to transform the limits under steady state conditions (Figures 2a, b) to those under non-steady state conditions (Figures 2c, d), and to better explain the calculation of the parameters alpha, beta and gamma.

**Referee#2: F. Jaramillo**

*The authors present a formulation for the use of the Budyko framework for non-steady conditions, i.e., with change in water storage within the basin. I find the manuscript an interesting approach that starts from definitions of water availability and energy demand in the "Turc space", later transposed to the "Budyko space", to end up with a formulation expressing the evaporative ratio in terms of change in storage and aridity index. Advantages: Their non-steady conditions formulation in its final way (Eq. 9) is simple, and can be obtained easily from any other steady-state formulation. It also confirms the robustness of Greve et al. (2016) and finds some important differences with those of Chen et al. (2013) and Du et al. (2013). I also appreciate the literature review on the theory behind the use of the Budyko framework for non-steady conditions.*

Thank you!

*Some suggestions to improve the manuscript are:*

*1. I find that the start from the "Turc space" and constant change to "Budyko space" gets confusing sometimes. Can't their formulation start directly from the much more commonly used "Budyko space"?*

We understand the questioning of the reviewer. Under steady-state conditions, the upper and lower limits are similar in both Turc and Budyko spaces. Moreover, both Turc-Mezentsev and Fu-Zhang functions, which are obtained from the resolution of a Pfaffian differential equation, have the following remarkable simple property: $B_1 = B_2$. However, this is not the

case for non-steady state conditions because the upper and lower limits differ when using the Turc or the Budyko space. The upper and lower limits and the transformation from steady to non-steady state conditions are easier to grasp in the Turc space. It is the reason why we prefer keeping both representations Turc and Budyko.

*2. I find the term S\* somehow difficult to grasp. First, why not just use DeltaS, for better clarity, instead of S=-delta S? Second, why not divide DeltaS (water) by P (water) instead of by EP (energy)? This would make much more sense, expressing the change in storage relative to P, something like S\*=DeltaS/P. I think in this way it would be so straightforward to use by anyone...*

We prefer keeping $S = - deltaS$ for two reasons: first to deal with a positive value when additional water is available for evapotranspiration and a negative value when water is withdrawn from precipitation; second to have a positive value which can be easily compared to the positive parameter $y_0$ of Greve et al.'s equation, one of the main results of the paper.

Thank you for this interesting suggestion to use $deltaS/P$. In fact, the adimensionalization of $S$ can either be made as $S^* = S/E_p$ or $S^{**} = S/P = (S/E_p)(E_p/P) = S^*Phi$ with $Phi = E_p/P$. All equations can be either written using $S^*$ or $S^{**}$, however the limits and the shape of the curves differ. The calculation can be easily made with $S^{**}$ and we will add the results in the revised version.

*3. The S limit definition of Line 12 page 3: 0<S<Ep, can the authors then explain in more detail this S limit definition (Line 12 page 3) for clarity? This because as it is, S is always positive, implying that delta S is always negative. So what about water storage in reservoirs (delta S >0), could the ML formulation for non-steady conditions also be used to represent this condition? Or if there is a typo there, could the ML formulation be applied conversely, deltaS<0, e.g. groundwater depletion for irrigation? See definition for both cases in "Local flow regulation and irrigation raise global human water consumption and footprint", 2015, Supplementary Information.*

Thank you again for this interesting suggestion. The methodology will be extended for negative values of $S$. We had already made the calculation and will add the results in the revised version. In the Turc and the Budyko spaces, the upper limits are similar for both cases $S > 0$ and $S < 0$, however the lower limits differ, and consequently the derivation of some equations will differ. The suggested reference will be cited.

*4. Upgrade the justification of their study (Line 20-21, page 2), something like a very-well needed validation, integration and comparison of non-steady formulations in Budyko space; that is what their work is from my point of view?*

Ok. A more detailed justification of the study will be added.

*5. Why would I prefer the ML formulation, please expand? I think the fact that noadditional parameters other than PET, P and deltaS to obtain ET/P for non-steady conditions is an important advantage.*

Ok. A more detailed explanation of the domain of application of the ML formulation will be added and discussed.

*6. One of the main conclusions is 25-28 page 8: Just by reading the corresponding discussion (Line 6-14, page 8) it is somehow difficult to understand. Can the authors use an additional figure comparing for the same storage conditions ALL the four formulations, Greve et al., ML, Chen et al. and Du et al.], either in the normal Budyko [Ep/P, E/P] or in the modified space [Ep/(P+dS), E/(P+dS)]. This synthesis would be very helpful for the reader and potential users of the ML formulation!*

Ok. We will add an additional figure comparing the four formulations, Greve et al., ML, Chen et al. and Du et al. in the modified space *[E$_p$/(P+dS), E/(P+dS)]* for both cases *S > 0* and *S < 0*.

---

## Author Response (AR2)

**Responses to reviewers**

We are very grateful to the Editor Dr Coenders-Gerrits, and to the reviewers, Drs Gudmundsson and Jaramillo, for their constructive comments of the manuscript. We totally agree with all their recommendations. Moreover, the title was shortened, the two sections 2 and 3 restructured, and Appendix A and the "Supplementary material" added. Corrections are in red in the revised manuscript.

**Editor Decision (19 Oct 2016): Publish subject to revisions (further review by Editor and Referees) by Miriam Coenders-Gerrits**

*Comments to the Author:*

*The authors changed their manuscript significantly. Although I think the revised version is better structured, I would like an assessment of the 2 reviewers as well since the recommended minor revisions are worked out as major revisions.*

We understand your decision because the suggestions of the reviewers opened new complementary and interesting perspectives.

*Furthermore, I highly recommend to change dS [L/T] into dS/dt. Storage is a stock with dimension [L] and thus has no time dimension. Although the authors define dS correctly in the list of symbols, it can become confusing after seeing Figure 1, where dS is defined as $S_e$-$S_b$ and where $S_e$ and $S_b$ are drawn with a line (indicating a storage level with dimension [L]).*

All terms *P, E, Q* and *ΔS* of the water balance equation ($P = E + Q + \Delta S$) have the same dimension [L/T] which represents a water depth (mm) during the same time period *Δt* as in Chen et al. (2013) and Du et al. (2016). In Figure 1, $\Delta S = S_e - S_b$ was substituted by $\Delta S = (S_e - S_b)/\Delta t$. The terms $S_b$ and $S_e$ are respectively the storage at the beginning and the end of the time period *Δt*.

**Editor Decision (12 Oct 2016): Publish subject to minor revisions (by Miriam Coenders-Gerrits)**

*The authors present an interesting study on a general formulation (ML) for the several Budyko curves that exist, especially for the non-steady state. The paper is generally well written and shows a good comparison of their ML-formulation and the existing equations.*

Thank you!

*The authors responded well to the questions and comments of the two reviewers and I think the new manuscript will be an improvement with sometimes a bit more clarification and additional analysis as suggested by the reviewers.*

Thank you. See below our responses to both reviewers.

*Additional minor comments from my side:*

*- Figure 1 might be skipped. I don't see the added value of it*

Ok, Figure 1 was skipped.

*- Be consistent with your symbols. Sometimes the storage change is named S, then dS. Besides, I think it will also be better to rename it to dS/dt, since it is storage change (delta) over time (dt). Once using dS/dt it is also more logical to have the dimension [L/T]. Normally S is the actual storage (dimension [L]).*

Ok, we totally agree because the notation $S = -\Delta S$ was confusing. The intermediate variable $S$ is no longer used, and equations were rewritten using only the storage change $\Delta S$. Moreover, in order to avoid confusion between the notations $S^*$ and $\Delta S$, we substitute $H_E$ to $S^*$ such as $H_E = S^* = -\Delta S/E_p$.

*- Maybe you can redefine the upper and lower boundary into $E_{max}$ and $E_{min}$ instead of $E_x$ and $E_n$.*

As equations and notations are very dense and complex in the whole manuscript, we prefer keeping $E_x$ instead of $E_{max}$ and $E_n$ instead of $E_{min}$ in order to simplify and reduce as possible the length of variables.

**Referee #1: L. Gudmundsson**

*The authors present an interesting study in which they propose an approach to extend Budyko functions to non-steady state conditions. The approach is based on a careful evaluation of the feasible limits of the Budyko-Turc space, which are subsequently adjusted for the case where additional water is available for evapotranspiration. This yields a general framework into which common Budyko functions can be inserted. Finally, the authors apply the proposed framework to popular Budyko functions and compare the results to previous studies.*

*Interestingly, the authors show that if their approach is applied to Fu's equation (Fu,1981), their approach yields an equation that is mathematically identical to a recent extension (Greve et al, 2016), after minor rearrangements. This finding increases the credibility of the results as the presented study and Greve et al (2016) have derived the same results on the basis of two independent approaches. Nevertheless, the presented work is clearly a new development as it (1) offers a more general approach that is applicable to a wide range of Budyko functions and (2) provides more explicit insights into the role of water storage (S) to the y0 parameter identified by Greve et al (2016).*

*Overall the paper is clearly structured and I find the graphical derivation of the proposed extension very convincing. Consequently, I do recommend the publication of the proposed work after some specific comments have been accounted for.*

Thank you!

*SPECIFIC COMMENTS:*

*1. ** ** Specific Comment 1: ** ***

*Although the paper is well structured I had the impression that it would benefit from some linguistic fine tuning and that some sections could presented more clearly.*

Ok. Additional comments were made to clarify some sections, and linguistic refinement to improve the text: i) the title was shortened "The Budyko functions under non-steady state conditions"; ii) all theoretical development of the new formulation were grouped in "Section 2" where two additional subsections 2.3 and 2.4 were added; iii) a subsection "3.3 Discussion", "Appendix" and "Supplementary material" were added; iv) in the whole text, many parts were rewritten (in red in the revised manuscript) and an additional schematic Figure 1 added.

*2. ** ** Specific Comment 2: ** ***

*The authors mention that the equation that is derived using their approach and Fu's equation, yields an equation that is "similar" to the equation derived by Greve et al (2016). In fact, the two equations are IDENTICAL after some minor re-arrangements, which is also shown by the authors. I therefore would like to urge the authors to clarify this issue in the revised manuscript. (As noted above, the authors work is nevertheless very valuable as it provides an independent validation of the previously derived function and allows for an explicit assessment of the amount of storage water that is available for evapotranspiration).*

Ok. The discussion comparing the ML and Greve et al's equations was clarified in order to discuss how two totally different methods give exactly the same result and equation (see Page 8, Lines 10, 27-28; Page 9, Lines 1-6, 22-28; Page 11, Lines 14-16). Moreover a comparison between the four formulations (ML, Greve, Chen et al., and Du et al.) was undertaken (see Page 10, Lines 2-7, 13-17, 27-29; Figure 7).

*3. ** ** Specific Comment 3: ** ***

*Page 4, lines 8ff: This section contains the actual derivation of the authors approach to incorporate storage water into Budyko functions. Unfortunately, I had to read this section several times before I could understand the logic underlying their approach. Therefore, I would like to encourage the authors to expand this section, and explain the important steps in more detail. More specifically: (1) I was wondering why the authors did search for the equation shown in line 12. (2) It took me a while to figure out how the values of beta, alpha and gamma were chosen (one or two sentences explaining the logic behind this would be helpful).*

Ok. This paragraph was totally rewritten in order to clarify the choice of the function used to transform the limits under steady state conditions (Figures 2a, b) to those under non-steady state conditions (Figures 2c, d and the new Figures 2e, 2f), and to better explain the calculation of the parameters alpha, beta and gamma (see Page 4, Lines 17-28; Page 5, Lines 1-26).

**Referee#2: F. Jaramillo**

*The authors present a formulation for the use of the Budyko framework for non-steady conditions, i.e., with change in water storage within the basin. I find the manuscript an interesting approach that starts from definitions of water availability and energy demand in the "Turc space", later transposed to the "Budyko space", to end up with a formulation expressing the evaporative ratio in terms of change in storage and aridity index. Advantages: Their non-steady conditions formulation in its final way (Eq. 9) is simple, and can be obtained easily from any other steady-state formulation. It also confirms the robustness of Greve et al. (2016) and finds some important differences with those of Chen et al. (2013) and Du et al. (2013). I also appreciate the literature review on the theory behind the use of the Budyko framework for non-steady conditions.*

Thank you!

*Some suggestions to improve the manuscript are:*

*1. I find that the start from the "Turc space" and constant change to "Budyko space" gets confusing sometimes. Can't their formulation start directly from the much more commonly used "Budyko space"?*

We understand the questioning of the reviewer. Under steady-state conditions, the upper and lower limits are similar in both Turc and Budyko spaces. Moreover, both Turc-Mezentsev and Fu-Zhang functions (obtained from the resolution of a Pfaffian differential equation) are identical in both spaces. However, this is not the case for non-steady state conditions because the upper and lower limits differ when using the Turc or the Budyko space. The upper and lower limits and the transformation from steady to non-steady state conditions are easier to grasp in the Turc space. It is the reason why we prefer keeping both representations Turc and Budyko. This discussion was added (see Page 4, Lines 18-27)

*2. I find the term S\* somehow difficult to grasp. First, why not just use DeltaS, for better clarity, instead of S=-delta S? Second, why not divide DeltaS (water) by P (water) instead of by EP (energy)? This would make much more sense, expressing the change in storage relative to P, something like S\*=DeltaS/P? I think in this way it would be so straightforward to use by anyone...*

We agree. As stated above, the intermediate variable $S$ is no longer used, and equations were rewritten using only the storage change $\Delta S$. However, we prefer keeping $H_E = S^* = -\Delta S/E_p$ for two reasons: first to deal with a positive value when additional water is available for evapotranspiration and a negative value when water is withdrawn from precipitation (see Page 3, Lines 20-28); second to have a positive value which can be easily compared to the positive parameter $y_0$ of Greve et al.'s equation, one of the main results of the paper (see Page 8, Lines 10-19).

Thank you for the interesting suggestion to use *deltaS/P*. In fact, the adimensionalization of *deltaS* can either be made as $H_E = -\Delta S/E_p$ or $H_P = -\Delta S/P = H_E \Phi$ (with $\Phi = E_p/P$). All equations were written using $H_E$ (in the main text) and $H_P$ (in Appendix A with the corresponding Figures and Tables in the "Supplementary material"). Both approaches were discussed (see Pages 7-8; Section 2.4).

*3. The S limit definition of Line 12 page 3: 0<S<Ep, can the authors then explain in more detail this S limit definition (Line 12 page 3) for clarity? This because as it is, S is always positive, implying that delta S is always negative. So what about water storage in reservoirs (delta S >0), could the ML formulation for non-steady conditions also be used to represent this condition? Or if there is a typo there, could the ML formulation be applied conversely, deltaS<0, e.g. groundwater depletion for irrigation? See definition for both cases in "Local flow regulation and irrigation raise global human water consumption and footprint", 2015, Supplementary Information.*

Thank you again for this interesting suggestion. The methodology was extended for negative values of $H_E$ and calculations added in the revised version. The corresponding Figures (1, 2, 3, 4, S1, S2 and S3) and Tables (S1, S2, S3 and S4) were modified. In the Turc and the Budyko spaces, the upper limits are similar for both cases $\Delta S < 0$ and $\Delta S > 0$, however the lower limits differ, and consequently the derivation of some equations differ. All equations, tables and figures distinguish now the two cases corresponding to $\Delta S$ negative and positive).

The suggested reference was cited (see Page 2, Line 12; Page 15, Lines 20-21).

*4. Upgrade the justification of their study (Line 20-21, page 2), something like a very-well needed validation, integration and comparison of non-steady formulations in Budyko space; that is what their work is from my point of view?*

Ok. A more detailed justification of the study was added (see Page 2, Lines 5-6, 12-14 and 21-26; Pages 7-8, section 2.4).

*5. Why would I prefer the ML formulation, please expand? I think the fact that no additional parameters other than PET, P and deltaS to obtain ET/P for non-steady conditions is an important advantage.*

Ok. A more detailed explanation of the domain of application of the ML formulation was added and discussed (see Page 11, Lines 1-10).

*6. One of the main conclusions is 25-28 page 8: Just by reading the corresponding discussion (Line 6-14, page 8) it is somehow difficult to understand. Can the authors use an additional figure comparing for the same storage conditions ALL the four formulations, Greve et al., ML, Chen et al. and Du et al.], either in the normal Budyko [Ep/P, E/P] or in the modified space [Ep/(P+dS), E/(P+dS)]. This synthesis would be very helpful for the reader and potential users of the ML formulation!*

Ok. We have modified Figure 7 comparing the four formulations, Greve et al., ML, Chen et al. and Du et al. in the modified space $[E_P/(P+dS), E/(P+dS)]$ (Page 10, section 3.3, Lines 10-29).

---

## Author Response (AR3)

**Responses to reviewers**

We are very grateful to the Editor Dr Coenders-Gerrits, and to the reviewers, Drs Gudmundsson and Jaramillo, for their constructive comments of the manuscript. We totally agree with all their recommendations.

**Editor Decision: Publish subject to technical corrections (16 Nov 2016) by Miriam Coenders-Gerrits**

Comments to the Author: Please have look at the minor comments of the 2 reviewers.

**Referee #1: L. Gudmundsson**

*I appreciate the authors revisions including the additional analysis which have been conducted based on suggestions made by the other reviewer. Overall it is still my evaluation that the authors do present an interesting and good trough thought analysis, which is now also presented more clearly. Admittedly, I did not have the time to check all the mathematical details, but I have carefully followed the logic of sections 2.1 and 2.2 as these forms the basis of the analysis. As these sections appear to be valid, I am confident that this is also the case for the remaining sections.*

*Apart from a few very minor suggestions listed below, I would fully support the publication of the presented paper in HESS.*

Thank you!

*Minor comments:*

*General: The jumps between the Turc and the Budyko space makes the article sometimes difficult to read. Therefore I would suggest to indicate which of the two options are used on a regular basis and at least once in each paragraph.*

Ok (see Page 3, Line 7; Page 4, Line 17; Page 5, Lines 2 and 16; Page 6, Lines 12, 17 and 27; Page 8, Line 15; Page 13, Lines 6, 7, 9, 10, 11 and 14; Page 18, Line 5).

*Abstract, line 10: DeltaS is referred to as change in soil water storage. Would it not be more correct to speak of changes in terrestrial water storage (which includes soils, groundwater, lakes, water stored in plants, snow, etc…)?*

Ok (see Page 1, Line 10; Page 2, Lines 11-12).

*Page 3, Line 7: On first reading it would be helpful if you include one sentence, stating that you focus on the Turc space as this is (in your evaluation) easier to grasp.*

Ok (see Page 3, Line 7).

*Page 5, lines 2-5: For me it still took a while to understand how you got to the three equations that specify the multipliers \alpha, \beta, \gamma. I would appreciate if you could walk through the three respective limits and explain each equation step by step.*

Ok, the text was rewritten and clarified (see Page 5, Lines 2-5).

*Page 5, line 4: The notation "\alpha . 0" is somewhat confusing. I assume you would like to indicate "alpha times zero". Maybe substituting the "." with a cross (x) may help?*

Ok, we substitute the "." with a cross "×" (see Page 5, Lines 3 and 18).

*Page 5, lines 15 – 18: See the two comments above, the same applies for this section.*

Ok, the text was rewritten and clarified (see Page 5, Lines 16-18).

*Page 6, line 15: change to "...the ML formulation in the Budyko space (Eqs 14a,b) ..." (see also first comment)*

Ok (see Page 6, Line 17).

*Page 8, lines 1 – 6: The reasons for the differences obtained for the He and Hp scaling are not 100% clear to me and I have difficulties to understand why one would be superior over the other. Could you please expand?*

Ok, the text was rewritten and clarified (see Page 8, Lines 3-9).

**Referee#2: F. Jaramillo**

*All is in the Recommendations to the Editor.*

Thank you!